# Influence of Connection Type and Platform Diameter on Titanium Dental Implants Fatigue: Non-Axial Loading Cyclic Test Analysis

**DOI:** 10.3390/ijerph17238988

**Published:** 2020-12-02

**Authors:** Ana I. Nicolas-Silvente, Eugenio Velasco-Ortega, Ivan Ortiz-Garcia, Alvaro Jimenez-Guerra, Loreto Monsalve-Guil, Raul Ayuso-Montero, Javier Gil, Jose Lopez-Lopez

**Affiliations:** 1Periodontal and Implant Surgery, CEIR Campus Mare Nostrum, School of Dentistry, University of Murcia, 30008 Murcia, Spain; ainicolas@um.es; 2Comprehensive Dentistry for Adults and Gerodontology, Faculty of Dentistry, University of Seville, 41009 Seville, Spain; ivanortizgarcia1000@hotmail.com (I.O.-G.); alopajanosas@hotmail.com (A.J.-G.); lomonsalve@hotmail.es (L.M.-G.); 3Faculty of Dentistry, University of Barcelona, 08007 Barcelona, Spain; raulayuso@ub.edu; 4Chairman of Bioengineering Institute of Technology, Universitat Internacional de Catalunya, 08017 Barcelona, Spain; xavier.gil@uic.es; 5Faculty of Dentistry, Service of the Medical-Surgical Area of Dentistry Hospital, University of Barcelona, 08007 Barcelona, Spain; jl.lopez@ub.edu

**Keywords:** connection type, dental implants, design, fatigue test, platform diameter

## Abstract

Two-pieces dental implants must provide stability of the implant-abutment-interface. The connection type and platform diameter could influence the biomechanical resistance and stress distribution. This study aims to evaluate the fatigue for different types of connections, external and internal, and different platform diameters. Three implant designs with the same length were used: (a) external hexagon/narrow platform; (b) internal double hexagon/narrow platform; (c) internal octagon/regular platform. A fatigue test was developed to establish the number of cycles needed before fracture. A 30º oblique load with a sinusoidal function of fatigue at a frequency of 15 Hz and 10% stress variation was applied to each system. The fatigue load limit (FLL) for design (a) was 190 N, being the nominal-curvature-moment (NCM) = 1.045; FLL = 150 N, with a NCM = 0.825 for (b), and FLL = 325 N, with a NCM = 1.788 for (c). The platform diameter affects the FLL, obtaining lower FLL on a narrow platform. The connection type interferes with the implant walls’ width, especially in narrow implants, making internal connections more unstable at this level. Long-term clinical studies to assess the restoration’s success rate and survival are mandatory.

## 1. Introduction

Two-piece dental titanium implants have been widely used for single-tooth replacements up to full-arch rehabilitation [1]. The implant-abutment-interface (IAI) [2] has to resist dynamic forces and be stable to withstand functional loads and to reduce screw loosing [3]. To maintain the stability of the IAI, different implant connection geometries have been developed, which can be summarized in two broad groups: external and internal connections. In terms of design, in the external connections, both the implant index and the prosthetic abutment index are located above the level of the implant platform. In contrast, in the internal connections, both structures are located inside the implant’s body, below the level of the implant platform [4].

The external connections are usually provided of an outer hexagon whose function is to provide rotational torque control during implant placement [5] and anti-rotational control between the implant index and the prosthetic abutment index. Several studies indicate that this type of connection is less favorable for stress distribution and has lower stability when compared to the internal connection [6].

The internal connection can present different designs depending on its geometric features and can be divided into an internal hexagon, internal octagon, trilobed system, or morse taper connection between others [7,8]. This type of design increases the implant-abutment contact area and improves the distribution and dissipation of forces, providing better stability [9], but is the internal conical connection (taper Morse connection) the one that shows the most intimate relation between the implant and the abutment, providing the most excellent stability and bacterial seal [10,11].

The presence of different design features (angles, channels, straight walls, cones, tubes) in the diverse connections types prevents rotation between the components of the system [12]. Its structural integrity is crucial for long-term stability [13], and some factors could induce deformation when the system is overloaded, over-torquing or non-axial forces are presented [14]. The thickness of the implant walls is a relevant factor since, sometimes, the design of the anti-rotational components inside the connection forces to leave walls excessively thin, especially in narrow implants [15,16].

The use of narrow implants is widely documented in patients with deficient bone crestal width in which, for some reason (increased healing time, cost, or patient morbidity), the application of horizontal bone regenerative techniques is not indicated [17,18,19]. The mechanical strength of titanium narrow implants is sometimes not enough to support the dynamic forces. The implant system does not offer long-term integrity of the connection complex, representing a significant risk of fractures [20,21]. Some aspects of the different connection configurations, such as biomechanical resistance and stress distribution, are crucial [22]. Not much is known about implant fatigue detailed by the type of connection. Hence, the objective of the present study was to evaluate the fatigue for different kinds of connections, external and internal, and different platform diameters, to establish which type of design supported higher values. Our null hypothesis was that indexation design and platform diameter influences titanium implant fatigue in the long-term.

## 2. Materials and Methods

### 2.1. Dental Implants

Fifty-four titanium dental implants from three different implant systems were compared in this study. The characteristics of each implant group are summarized in Table 1 and each implant design is exposed in Figure 1:-Group I (n = 19): Surgimplant CE: titanium grade 5 dental implant with hexagon external connection (platform: 3.5 mm, length: 12 mm) (Galimplant SLU, Sarria, Lugo, Spain)-Group II (n = 18): Surgimplant CI Double Hexagon: titanium grade 5 dental implant with double hexagon internal connection (platform: 3.5 mm, length: 12 mm) (Galimplant SLU, Sarria, Lugo, Spain)-Group III (n = 17): Surgimplant CI Octagonal: titanium grade 5 dental implant with octagonal internal connection (platform: 4.0 mm, length: 12 mm) (Galimplant SLU, Sarria, Lugo, Spain).

### 2.2. Fatigue Test

A fatigue test was performed to obtain the number of cycles before fracture. The maximum and minimum force applied was recorded for each sample. The assays were performed with a servo-hydraulic testing machine (MTS 858 Mini Bionix II, MTS, Minneapolis, MN, USA) equipped with a load cell MTS 661.19F-01 of 5 kN.

The spherical member of the load application was made of titanium grade 5 (Figure 2).

The implants were fixed 30° angulated with the axis z of the load cell (Figure 3). They were loaded with a sinusoidal function of fatigue at a frequency of 15 Hz and 10% stress variation. The error during loading measurements was less than 5 N, and the maximum loading applied to the implant was around 80% of the value of the implant failure load, obtained by a static test under the same geometric conditions as fatigue tests, following ISO 14801:2008 recommendations [23]. All tests were carried out under stable environmental conditions with a temperature of 25 °C and relative humidity of 60%.

### 2.3. Scanning Electron Microscopy (SEM) Analysis

The fracture samples were observed by SEM at 10 kV using a Neon 40 Focused Ion Beam Scanning Electron (FIB-SEM) microscope (Carl Zeiss NTS GmbH, Oberkochen, Germany).

### 2.4. Statistical Analysis

Statistically significant differences among the three groups were assessed using SPSS 18.0 software (SPSS Inc., Chicago, IL, USA). Differences between groups were analyzed by Analysis of Variance (ANOVA), and a multiple comparison Fisher test was applied. The level of significance was established at a *p*-value of 0.05.

## 3. Results

The failure mode was similar in all experimental groups, including large deformations at the implant neck area. The implant neck fracture took place most of the cases between the first and second threads.

### 3.1. Hexagon External Connection

The minimum and maximum load expressed in Newtons (N) applied to all the samples of the hexagon external connection group was 190 N and 400 N, respectively. The cycles applied before fracture were between 3074 and 5,000,000. The cyclic load diagram obtained from the results of the test is shown in Figure 4.

The fatigue load limit (F_FL_, according to ISO 14801:2008) was F_FL_ = 190 N, being the nominal curvature moment (N.m) = 1.045.

### 3.2. Double Hexagon Internal Connection

The minimum and maximum load expressed in Newtons (N) applied to all the samples of the double hexagon internal connection group was 150 N and 400 N, respectively. The cycles applied before fracture were between 1583 and 5,000,000. The cyclic load diagram obtained from the results of the test is shown in Figure 5.

The fatigue load limit (F_FL_, according to ISO 14801:2008) was F_FL_ = 150 N, being the nominal curvature moment (N.m) = 0.825.

### 3.3. Octagonal Internal Connection

The minimum and maximum load expressed in Newtons (N) applied to all the samples of the octagonal internal connection group and the cycles applied before fracture were 325 N and 550 N respectively. The cycles applied before fracture were between 3555 and 5,000,000. The cyclic load diagram obtained from the results of the test is shown in Figure 6.

The fatigue load limit (F_FL_, according to ISO 14801:2008) was F_FL_ = 325 N, being the nominal curvature moment (N.m) = 1.788.

A summary of the results of the three experimental groups is shown in Table 2.

The lack of retention between the abutment and dental implant was assessed as a failure. The fracture mechanism starts by abutment screw loosening producing cracks on the surface that grow with the load cycles and later fracture, but not due to destruction of the implant neck or shoulder. Analysis of fractured screws by SEM revealed that the mode and the region of fracture were the same for the different systems studied. The fracture surfaces were similar for all implants corresponding to the connection zone and fractured the body of the implant, according to the indications of the international standards for fracture fatigue behavior [24] for the dental implants with good mechanical behavior.

Statistically, the hexagonal external connection presented a lower fatigue limit load with statistical differences significance than the double hexagonal internal connection (*p* < 0.012) and also in relation to the octagonal internal connection (*p* < 0.003). When both internal connections are compared, the octagonal connection presents a higher limit fatigue load than the double hexagonal connection with statistical differences significance (*p* < 0.004).

The striations from the fractography can be observed in Figure 7, where the crack grows from the surface specimens and from 10-mm beneath the surface. In all cases, we observed the same morphology of fracture. The equiaxed grains can be observed, and in their faces, the marks of the crack in the propagation process to fracture.

## 4. Discussion

This experimental study aimed to evaluate the influence of the connection features and platform diameter in the fatigue response of titanium grade 5 dental implants. Three different implant systems were assessed, one narrow implant system (3.5 mm platform) with an external hexagonal connection, one narrow implant system (3.5 mm platform) with double hexagon internal connection, and one regular platform (4.0 mm) implant system with an octagonal internal connection.

The implant-abutment interface geometry is an influencing factor for the transmission of stress around the implant [25].

This experimental test is a reliable method to determine the effect of different parameters, such as connection design or platform diameter, on implant dynamic failure strength. The same company manufactured the three experimental groups and the three spherical members attached to the connection, using the same titanium grade 5. This fact is one of the strengths of the present study. In most studies, authors compare failure strength between different implants and different abutment interfaces, with different shapes, surface characteristics, and material properties, and the comparability is compromised [26].

In addition to the factors mentioned above, some factors inherent in the host may affect the distribution of strain and stress in bone and implants. A study developed by Oliveira et al. concluded that the density of the medullar bone and the thickness of the cortical bone also affect the distribution of strain and stress, negatively affecting the decrease in medullar bone density [27].

Lo Giudice et al. demonstrated that the bone preparation could also affect the bone quality showing better results in osteotomies performed with ultrasonic tips and concluding that the use of the piezosurgery preserves the bone morphology and decreases the presence of microfractures [28]. The marginal bone loss around implants is also influenced by the fact the implant is placed in native bone or placed in grafted tissues. Galindo-Moreno et al. found in a retrospective cohort study that implants placed in grafted tissues showed more marginal bone loss than implants placed in pristine bone [29]. The type of connection also affects the marginal bone stability, being the external connections strongly associated with an increased marginal bone loss, not only in the first twelve months but over time [29].

Also, bone quality and crestal bone loss can be influenced by other factors. Not only does the neck shape, microthreads, or surface texture affect crestal bone stability, but the implant-abutment connection appears to be a significant factor on peri-implant crestal bone level [30]. The abutment height also has an important role, as demonstrated in an in vivo study developed by Spinato et al. They suggested that the shorter the abutment height, the greater the marginal bone loss, especially in cement-retained prosthesis [31].

Several studies have tested dental implants using static loading, while others use cyclic loads [32]. Most of the reviews focus on implant design but does not mention fatigue as a complex failure mode [22]. A few studies have considered the effect of the implant diameter on fatigue performance, concluding that narrow implants failed to show typical fatigue behavior, which might be attributed to the implant design [33]. The inconsistent fatigue behavior observed for narrow implants could result from factors like notches, dents, or machining markings. To date, no studies have been published about a fracture mode analysis to support this assumption [33].

Carneiro et al. [34] developed an in vitro study evaluating the fracture resistance of internal and external hexagon in regular and narrow implants, concluding that titanium is a material that presents no clear evidence of the exact point between the plastic and elastic limits. No significant reduction of the blending elastic limit was found between narrow and regular internal connections.

In our study, the failure due to the bending elastic limit was observed at 190 N with external hexagon narrow implants versus the 150 N resulted in the internal hexagon narrow implants. This result could be because, in the internal connection, the indexation features are ubicated inside the implant’s body, leaving thinner walls than external connections. Besides, an important cause of the high fatigue life of the octagonal internal connection is the size of the resistant section. The double hexagonal internal connection and external system present a higher value of the area than the internal.

Our test simulated the clinical situation when the stress concentration resulting from occlusal forces leads to microfractures and bone loss around the implant, leading to mobility and fracture of the implant [35,36]. On the other hand, our results showed an increase from 150 N to 325 N in the elastic limit by increasing the platform diameter in the internal connection from 3.5 to 4 mm. This difference between the platform diameter was not significant in the results showed by Carneiro et al., although they found a more substantial number of cracks in the narrow implants than the regular.

Tolerances of manufacturing are the main reason for the loose-fit of the components and required the manufacturer to improve the fit. In these situations (loose-fit), the possibility of horizontal movement and rotation between screw and implant and lower than the forces to tighten it, micromovements could have led to a progressive unscrewing of the abutment screw under conditions dynamic loading. The most cause of the high fatigue life of the external connection is the size of the resistant section. The external system presents a lower value of the surface than the internal. This fact produces a worse load distribution. This reason explains the differences in mechanical properties. Besides, the tolerances in the internal connections are better, and this good finishing provokes a higher fatigue limit of the internal connection system [37,38,39].

Each implant-abutment interface has its advantages and disadvantages. According to Maeda et al. [40], the external hexagon interface has benefits such as suitability for the two-stage method, provision of an anti-rotation mechanism, retrievability, and compatibility among different systems. However, increased screw loosening, component fracture, and difficulty in seating abutments in deep subgingival tissues are problems commonly experienced with external hexagon connectors [41].

The advantages of the internal hexagon following Maeda are ease in abutment connection, suitability for one stage implant installation, higher stability and suitability for single-tooth restoration, higher resistance to lateral loads due to the lower center of rotation, and better force distribution.

The masticatory loading at anterior regions is variable with a mean value of 286 N, s.d. 164 N, while the posterior area shows a mean value of 579 N, s.d. 235 N [42]. Those data showed a high subject variability so that the use of narrow implants is recommended just for the anterior region. In the posterior area, it is mandatory to use a wider platform.

The present study results support the acceptance of the null hypothesis tested since there was a difference in the maximum force supported in narrow implants (internal or external connections) and regular platform implants. Different types of connections also presented differences in the fatigue load limit. Clinical studies are mandatory to test the stability of the different connections evaluated, assessing the success rate and survival of the prosthesis in anterior and posterior teeth.

## 5. Conclusions

Based on the present study results and within the limitations of the same, we may conclude that the platform diameter affects the fatigue load limit, obtaining a lower fatigue load limit implants with the narrow platform (3.5 mm) than the regular platform (4 mm). On the other hand, the indexation design may interfere with the width of the implant walls, especially in narrow implants, making internal connections more unstable at this level. It would be advisable to develop long-term clinical studies to assess the restoration’s success rate and survival.

## Figures and Tables

**Figure 1 ijerph-17-08988-f001:**
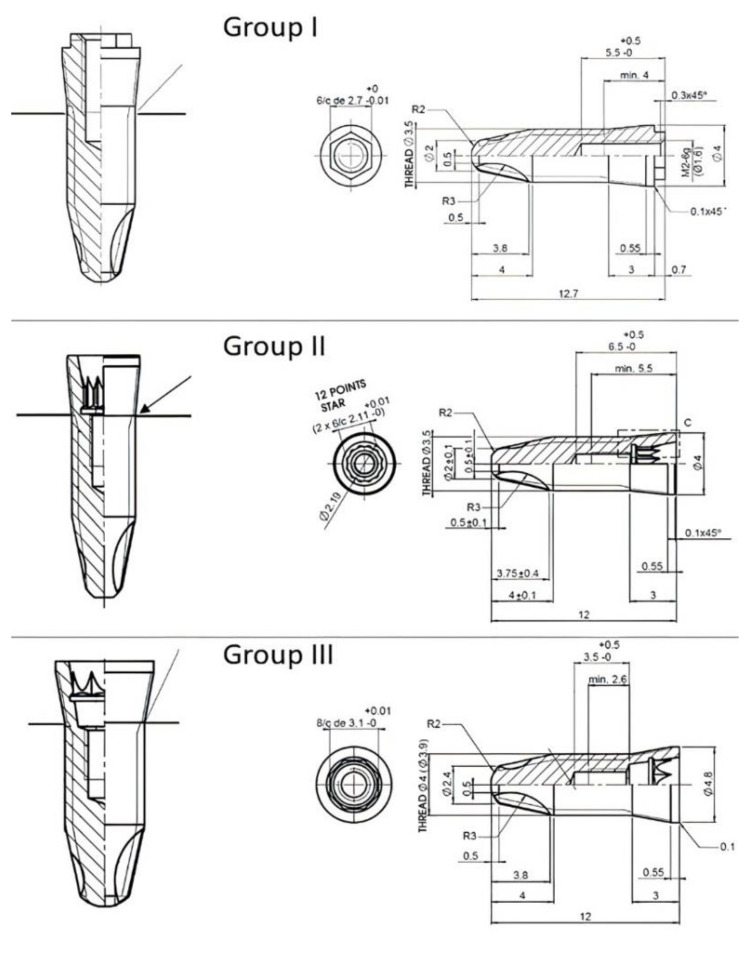
Implant design of each experimental group.

**Figure 2 ijerph-17-08988-f002:**
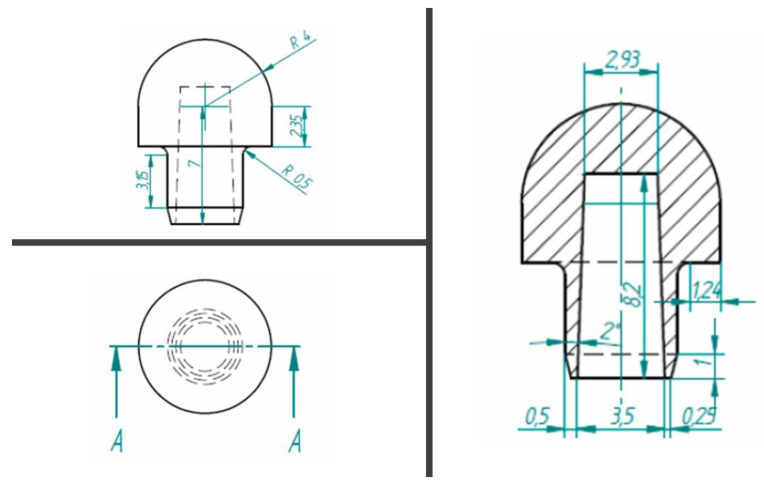
Spherical member of the load application design details.

**Figure 3 ijerph-17-08988-f003:**
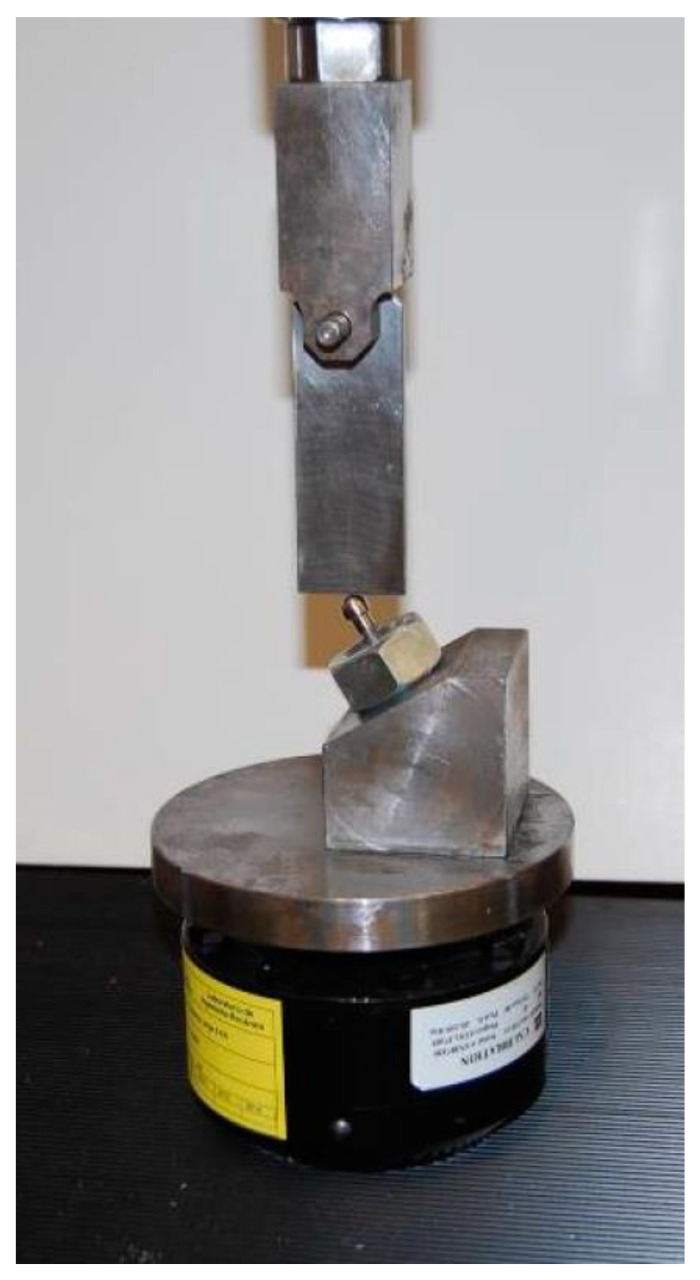
Load cell over the sample in the testing machine.

**Figure 4 ijerph-17-08988-f004:**
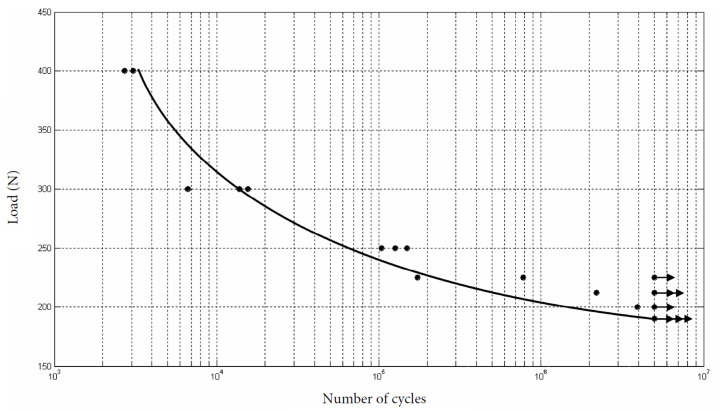
Cyclic load diagram for hexagon external connection obtained from the results of the test showed in Table 2.

**Figure 5 ijerph-17-08988-f005:**
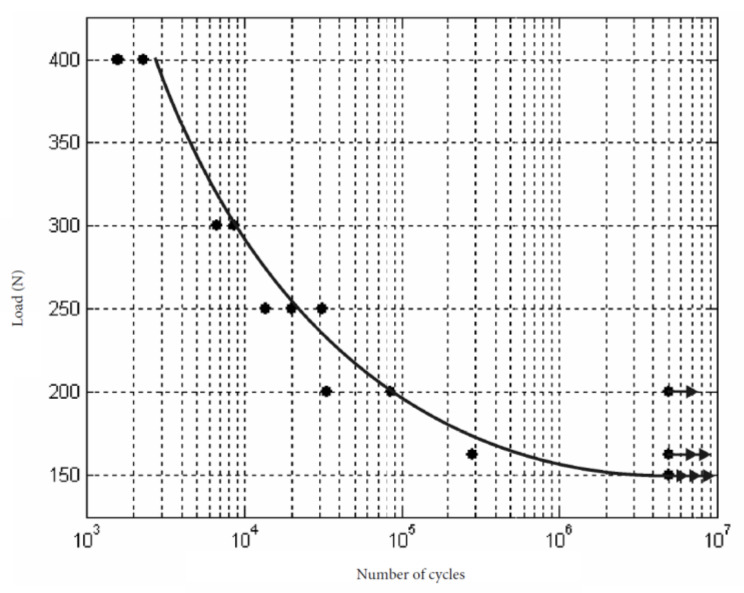
Cyclic load diagram for double hexagon internal connection group obtained from the results of the test.

**Figure 6 ijerph-17-08988-f006:**
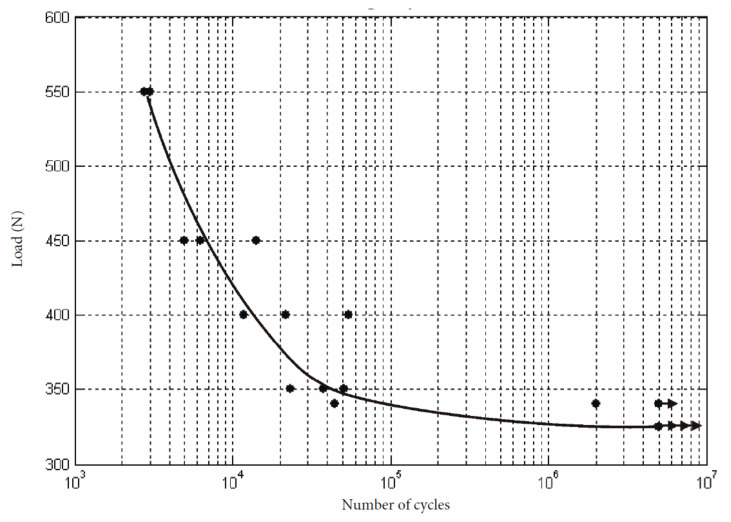
Cyclic load diagram for the octagonal internal connection group obtained from the results of the test.

**Figure 7 ijerph-17-08988-f007:**
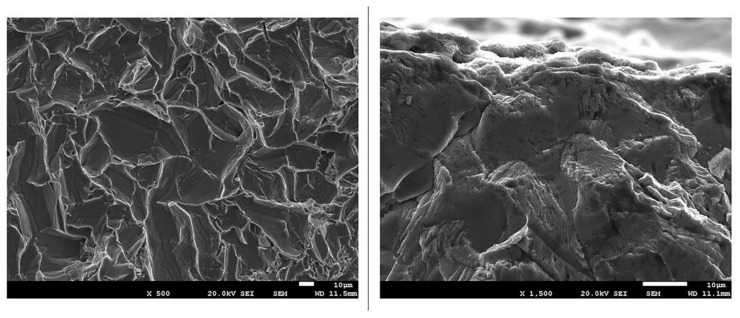
SEM images at a magnification of ×500 and ×1500 showing the striations from the fractography.

**Table 1 ijerph-17-08988-t001:** Implant characteristics distributed by groups.

Group	Group I	Group II	Group III
n	19	18	17
Connection Type	Hexagon External Connection	Double Hexagon Internal Connection	Octagonal Internal Connection
Diameter	3.5	3.5	4.0
Length	12	12	12
Material	Titanium Grade 5	Titanium Grade 5	Titanium Grade 5

**Table 2 ijerph-17-08988-t002:** Summary of the results obtained in each experimental group.

Implant Type	Minimum Load (N)	Maximum Load (L)	Minimum Cycles	Maximum Cycles	Fatigue Load Limit (F_FL_) (N)	Nominal Curvature Moment (N.m)
Hexagon external connection	190	400	3074	5,000,000	190	1.045
Double hexagon internal connection	150	400	1583	5,000,000	150	0.825
Octagonal internal connection	325	550	3555	5,000,000	325	1.788

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
