# Peer review of "Influence of Connection Type and Platform Diameter on Titanium Dental Implants Fatigue: Non-Axial Loading Cyclic Test Analysis"

_ijerph, 2020, doi:10.3390/ijerph17238988_

Round 1

Reviewer 1 Report

The article is interesting but some aspect need to be clarified
The English language in the whole text appears too colloquial and few grammar errors are present, it need to be revised (ex. line 32-44, 55,69 ,94,178 ) Introduction The introduction provide the correct background. Materials and Method Please revise line 94 The authors need to perform the power analysis to decide an appropriate sample population. In line 157 the A.A. discuss about a sem analysis performed on samples; how this evaluation was performed is not reported in material and method. I suggested add some picture and to discuss this findings in the Discussion section.

Translate from spanish to english in fig 1 

Regarding the bone characteristics that may affect the stress distribution it could be useful for the readers to understand if the bone preparation could affect the bone quality. (line 182-185) Lo Giudice R, Puleio F, Rizzo D, Alibrandi A, Lo Giudice G, Centofanti A, et al. Comparative investigation of cutting devices on bone blocks: An SEM morphological analysis. Appl Sci 2019;9(2). 

In discussion section please rewrite lines 199-203   /207-218 to be more clear

It is suggested to write a conclusive paragraph 

Author Response

Dear Reviewer,

Thank you for your time and comments that positively improve the quality and clarity of our work.

Following your recommendations, the following modifications were completed:

  1. The authors need to perform the power analysis to decide an appropriate sample population.

In the fatigue tests, the samples are used to obtain the S-N curve with enough points to see their asymptotic function and to be able to determine the fatigue limit. In our contribution, 54 implants have been used in total for the three different systems: 19, 18 and 17 for each of them. It can be observed by the results obtained that they are sufficient to obtain very clear results of the behavior to fatigue. Sometimes, it happens that there is great disparity of results but in the three cases studied the samples have fractured in the number of cycles that allow determining their behavior. Therefore, the samples used are more than sufficient.

  1. In line 157 the A.A. discuss about a sem analysis performed on samples; how this evaluation was performed is not reported in material and method.

A new paragraph (2.3. Scanning Electron Microscopy (SEM) Analysis) has been added to material and methods describing the MEB evaluation.

  1. I suggested add some picture and to discuss this findings in the Discussion section.

A new Figure (7) has been added with SEM images of the striations from the fractography.

  1. Translate from spanish to english in fig 1 

Translation has been done.

  1. Regarding the bone characteristics that may affect the stress distribution it could be useful for the readers to understand if the bone preparation could affect the bone quality. (line 182-185) Lo Giudice R, Puleio F, Rizzo D, Alibrandi A, Lo Giudice G, Centofanti A, et al. Comparative investigation of cutting devices on bone blocks: An SEM morphological analysis. Appl Sci 2019;9(2). 

A new paragraph has been added to the discussion section and the cite has been added to the bibliography as suggested.

  1. It is suggested to write a conclusive paragraph 

The conclusive paragraph appears at the end of the discussion section: “The present study results support the acceptance of the null hypothesis tested since there was a difference in the maximum force supported in narrow implants (internal or external connections) and regular platform implants. Different types of connections also presented differences in the fatigue load limit. Clinical studies are mandatory to test the stability of the different connections evaluated, assessing the success rate and survival of the prosthesis in anterior and posterior teeth.”

Reviewer 2 Report

The authors compared three implant designs using fatigue tests. The experimental is well designed and the conclusion could be supported by the results. There are some concerns to be considered before publication:

The statistical analysis was not well described in the methods. Did the authors compare the maximum load for all the cycles passed 5000000? For example, did the authors use 225, 212, 212, 200, 190, 190, 190 for Group A?

The tables and figures have duplicated information. The authors might remove all tables, but add one figure or table to statistically compare the three groups.

There are minor typos needed to be corrected over the whole manuscript.

Author Response

Dear Reviewer,

Thank you for your time and comments that positively improve the quality and clarity of our work.

Following your recommendations, the following modifications were completed:

  1. The statistical analysis was not well described in the methods. Did the authors compare the maximum load for all the cycles passed 5000000? For example, did the authors use 225, 212, 212, 200, 190, 190, 190 for Group A?

The statistical studies have been carried out with the 54 implants analyzed. However, since we have the problem of the different loads used in the fatigue studies, the same load is always used. In some cases, the experimental values are used, and in those cases where we do not have the experimental value, it is obtained from the S-N curve obtained. In fact, this fact can be considered a limitation of the present study, but it is a technique used by other different authors in fatigue studies:

  • Semlitsch MF, Panic B, Weber H and Schoen R. Comparison of the fatigue strength of femoral prosthesis stems made of forged Ti-Al-V and Cobalt base alloys. In “Titanium Alloys in Surgical Implants. ASTM-STP796”. Luckey and Kubli, eds; ASTM, Philadelphia; 1981, p.120-135.
  • Manero JM, Gil FJ and Planell JA. Effect of saline solution environment on the cyclic deformation of Ti-6Al-4V”. J Mater Sci: Mat Med, 1996; 7:131-134.
  • Svehla M, Morberg P, Zicat B, Bruce W, Sonnabend D and Walsh WR. Morphometric and mechanical evaluation of titanium implant integration: Comparison of five surface structures. J Biomed Mater Res, 2000; 51:15-22.

  1. The tables and figures have duplicated information. The authors might remove all tables, but add one figure or table to statistically compare the three groups.

Tables 2 to 4 have been eliminated. A new table (“Table 2”) has been added with the summary of the results.

Reviewer 3 Report

Review on Nicolas-Silvente et al’ „Influence of Connection Type and Platform Diameter on Titanium Dental Implants Fatigue: Non-Axial Loading Cyclic Test Analysis “

Introduction: Contains and clearly describes all important information targeting the hypothesis.

Materials and methods: Adequate methods used. Figures illustrate the content properly.

Results: Clear and sufficient.

Discussion: Compares all results with previous findings.

Just there is one misstyping in line 188:…considerer…must be changed to considered!

Author Response

Dear Reviewer,

Thank you for your time and comments that positively improve the quality and clarity of our work.

Following your recommendations, the following modifications were completed:

Discussion: Compares all results with previous findings.

Comparison has been made with the author cited [29, 30, 31, 32,33], lines 236 to 258

Just there is one misstyping in line 188:…considerer…must be changed to considered!

Thank you for reporting the mistyping.

Reviewer 4 Report

Both in the Introduction and in the Discussion the Authors should briefly analyze the Influence of Connection Type and Platform Diameter on crestal bone loss "in vivo" (some clinical studies can be cited, for example:

Spinato S, Galindo-Moreno P, Bernardello F, Zaffe D. Min- imum abutment height to eliminate bone loss: In uence of implant neck design and platform switching. Int J Oral Maxillofac Implants 2018;33:405–411.

Penarrocha-Diago, M. A., et al. Influence of implant neck design and implant-abutment connection type on peri-implant health. Radiological study. Clinical oral implants research. 2012.

Galindo-Moreno, P., Fernandez-Jimenez, A., Avila-Ortiz, G., Silvestre, F. J., Hernandez-Cortes, P. & Wang, H. L. Marginal bone loss around implants placed in maxillary native bone or grafted sinuses: a retrospective cohort study. Clinical oral implants research. 2014.

In the Discussion clinical advantages /disadvantages of external/ internal connection should be briefly exposed

Author Response

Dear Reviewer,

Thank you for your time and comments that positively improve the quality and clarity of our work.

Following your recommendations, the following modifications were completed:

  1. Both in the Introduction and in the Discussion the Authors should briefly analyze the Influence of Connection Type and Platform Diameter on crestal bone loss "in vivo" (some clinical studies can be cited, for example:

Spinato S, Galindo-Moreno P, Bernardello F, Zaffe D. Minimum abutment height to eliminate bone loss: Infuence of implant neck design and platform switching. Int J Oral Maxillofac Implants 2018;33:405–411.

Penarrocha-Diago, M. A., et al. Influence of implant neck design and implant-abutment connection type on peri-implant health. Radiological study. Clinical oral implants research. 2012.

Galindo-Moreno, P., Fernandez-Jimenez, A., Avila-Ortiz, G., Silvestre, F. J., Hernandez-Cortes, P. & Wang, H. L. Marginal bone loss around implants placed in maxillary native bone or grafted sinuses: a retrospective cohort study. Clinical oral implants research. 2014.

The references recommended have been included in the discussion section.

  1. In the Discussion clinical advantages /disadvantages of external/ internal connection should be briefly exposed

A new paragraph was added (lines 238-240) discussing external vs. internal connections.

Round 2

Reviewer 2 Report

The authors replied that they have combined the tables to one table. Thanks for sending the revised version of the manuscript. The changes are acceptable and there are no need for further changes.